# Indirect Chronic Effects of an Oleuropein-Rich Olive Leaf Extract on Sucrase-Isomaltase In Vitro and In Vivo

**DOI:** 10.3390/nu11071505

**Published:** 2019-07-01

**Authors:** Alison Pyner, Shuk Yan Chan, Sarka Tumova, Asimina Kerimi, Gary Williamson

**Affiliations:** 1School of Food Science and Nutrition, University of Leeds, Leeds LS2 9JT, UK; 2Department of Nutrition, Dietetics and Food, School of Clinical Sciences at Monash Health, Faculty of Medicine, Nursing and Health Sciences, Monash University, Notting Hill BASE facility, 264 Ferntree Gully Road, Notting Hill, VIC 3168, Australia

**Keywords:** Oleuropein, olive leaf extract, sucrase, Caco-2, diabetes, sugar

## Abstract

Consumption of dietary bioactives is an avenue to enhancing the effective healthiness of diets by attenuating the glycaemic response. The intestinal brush border enzyme sucrase-isomaltase (SI) is the sole enzyme hydrolysing consumed sucrose, and we previously showed the acute effects of olive leaf extract (OLE) on sucrase activity when given together with sugars both in vitro and in vivo. Here we tested whether OLE could affect sucrase expression when pre-incubated chronically, a “priming” effect not dependent on competitive interaction with SI, in both a cell model and a human intervention. Using differentiated Caco-2/TC7 cells, long-term pre-treatment with oleuropein-rich olive leaf extract (OLE) lowered SI mRNA, surface protein and activity, and attenuated subsequent sucrose hydrolysis. Based on these results, a randomised, double-blinded, placebo-controlled, crossover pilot study was conducted. OLE (50 mg oleuropein) was consumed in capsule form 3 times a day for 1 week by 11 healthy young women followed by an oral sucrose tolerance test in the absence of OLE. However this treatment, compared to placebo, did not induce a change in post-prandial blood glucose maximum concentration (Glc_max_), time to reach Glc_max_ and incremental area under the curve. These results indicate that changes in SI mRNA, protein and activity in an intestinal cell model by OLE are not sufficient under these conditions to induce a functional effect in vivo in healthy volunteers.

## 1. Introduction

Food components which lower the glycaemic response of carbohydrate-rich meals could help attenuate the overall dietary glycaemic load and potentially decrease the risk of type 2 diabetes [1]. We have previously shown an acute activity for pomegranate and for mixed fruit puree with bread [2,3], and others have shown comparable effects for cinnamon on rice [4], for mixed berry puree with sucrose [5], and for apple extract with glucose [6]. The proposed underlying mechanisms are inhibition of the starch digesting enzymes, α-amylase and α-glucosidase [7], and effects on sugar transport via intestinal [8] or kidney [6] sugar transporters.

The health benefits of olives and olive oil are well documented, but although originally attributed to the constituent oleic acid, the complementary effects of the (poly)phenolic components have been recognized [9,10]. The key (poly)phenols found in olives (*Olea europaea*) are the secoiridoid oleuropein and its derivatives hydroxytyrosol and tyrosol. Olive leaves have a higher concentration of oleuropein compared to the olive fruit [11] while an olive leaf extract (OLE) has been reported to have several functional properties [10,12]. We have shown that oleuropein inhibits acutely sucrase activity when sucrose is directly in combination with OLE, and that this leads to reduced plasma glucose following a dose of sucrose when consumed together with OLE in healthy volunteers [13]. The acute effect on sucrose hydrolysis is independent of oleuropein absorption, since SI is a brush border enzyme and hence in contact with the contents of the gut lumen. Oleuropein in the intact form is absorbed from the intestine after consumption with about 0.9 μM in plasma after consumption of oleuropein capsules containing 77 mg oleuropein [14], together with several metabolites such as sulfated and glucuronidated hydroxytyrosol [14]. Furthermore, chronic intake of OLE inhibited the progression of type 2 diabetes in mice [15]. In humans, a double-blind crossover study with 6-week consumption of OLE led to a 15% improvement in insulin sensitivity, a 6% and 14% reduction in fasting blood glucose and fasting insulin respectively [16]. Here we show the effects of an oleuropein-rich OLE on sucrase-isomaltase (SI) expression in Caco2/TC7 cells, and the chronic effect of OLE on postprandial blood glucose following sucrose consumption in the absence of OLE. 

## 2. Material and Methods

### 2.1. Caco-2/TC7 Cell Culture

Caco-2/TC7 cells were a kind gift from Prof. Monique Rousset, Centre de Recherche des Cordeliers, Paris, France, and were routinely cultured at a density of 1.6 × 10^4^ cells/ cm^2^ in T75 cell culture flasks in glucose-free DMEM medium, 2% (*v*/*v*) Glutamax (Gibco, Thermo Fisher Scientific, Waltham, MA, USA) supplemented with either 25 mM glucose or 25 mM sucrose, 20% (v/v) foetal bovine serum, 2% (*v*/*v*) non-essential amino acids and 1% penicillin-streptomycin solution (all from Sigma-Aldrich, Dorset, UK). Cells were maintained at 37 °C with 10% CO_2_ and sub-cultured by detaching with 0.25% trypsin at 80% confluence after 3 d. Cells were seeded at 6.4 × 10^4^ cells/ cm^2^ on solid or Transwell supports and allowed to fully differentiate for 21 d (Corning, N., USA). 

### 2.2. Chronic OLE Treatment

OLE (Olecol^®^ in capsules and Bonolive^®^ as a loose powder) was kindly supplied by BioActor B.V. (Maastricht, The Netherlands) containing 19.8 ± 0.5% and 41.8 ± 0.9% (*w*/*w*) of oleuropein respectively [13]. For chronic OLE treatment, cells were exposed to OLE as Bonolive for the final 3 d of differentiation (days 18–20) and the assay(s) were conducted on day 21. Time course and concentration response experiments were prepared as detailed in figure legends. Other components in the OLE were inert excipients added for encapsulation. By HPLC, Bonolive^®^ powder also contained low amounts of tyrosol (0.045% *w*/*w*) and hydroxytyrosol (0.0065% *w*/*w*) as previously reported [13]. Caco-2/TC7 cell viability was determined for experiments after 3-day treatment with OLE at 1.5 mg/mL. Cells were seeded on 12-well solid supports and maintained and treated as previously described. Viability was determined using the LDH cytotoxicity detection kit Roche, Burgess Hill, UK according to the manufacturer’s protocol. No significant loss of viability was observed (*n* = 9 replicates).

### 2.3. SI Activity Assay

Differentiated cells in 75 cm^2^ flasks were washed 3 times with cold PBS and scraped into 1 mL of 0.1 M phosphate buffer pH 7.0 containing 4-(2-aminoethyl)benzenesulfonyl fluoride hydrochloride, aprotinin, bestatin hydrochloride, N-(trans-epoxysuccinyl)-L-leucine 4-guanidinobutylamide, leupeptin hemisulfate salt, and pepstatin A as protease inhibitors (Sigma-Aldrich). The cell lysates were snap frozen and stored at −80 °C until required. On the day of the assay, cell lysates were thawed, vortexed and then passed 10–15 times though a 21G needle syringe. After protein determination [17], the lysate was diluted in assay buffer as required. Assay samples of 250 µL were used containing 0.02 units of sucrase activity; where 1 unit is the amount of enzyme required to produce 1 µmol of product per min. An incubation time of 10 min was used to fall within the linear range of product with 10 mM sucrose as substrate. To determine the specific activity, four concentrations of enzyme were tested. The product glucose concentration was determined using a hexokinase assay. The assay has been as described in full previously [18]. Absorbance measurements were performed on a Pherastar FS plate reader (BMG Labtech, Germany).

### 2.4. Droplet digital PCR Analysis of SI, GLUT2, GLUT5, SGLT1 and CDX2 

Caco-2/TC7 cells were seeded on 6-well culture plates and grown as described in *2.1* and treated with OLE as described in 2.2. On day 21, the cells were washed with ice-cold PBS and total RNA was extracted using the RNAqueous kit (Ambion, Life Technologies, Thermo Fisher Scientific). RNA content was determined spectrophotometrically at 260 nm on a NanoDrop (Thermo Fisher Scientific) and cDNA was synthesised from 250 ng of total RNA using the GoScript Reverse Transcription System (Promega, Madison, WI, USA). The QX100 Droplet Digital PCR system (ddPCR; Bio-Rad Laboratories, CA, USA) was used to quantify changes in gene expression of SI, GLUT2, GLUT5, SGLT1 and CDX2 from *n* = 9 replicates. Each assay (20 µL) consisted of cDNA template (2.5 ng for SI, CDX2 or 5 ng for GLUT2, GLUT5 and SGLT1), 1 µL of FAM-labelled Taqman primer for each gene of interest, 1 µL of VIC-labelled Taqman primer for TBP (TATA-box binding protein, housekeeping gene) and 10 µL of ddPCR Supermix (Bio-Rad Laboratories). The Taqman primers were from Life Technologies (Thermo Fisher Scientific) Hs00356112_m1 (SI), Hs01096905_m1 (GLUT2) Hs01086390_m1 (GLUT5), Hs01573790_m1 (SGLT1), Hs01078080_m1 (CDX2) and Hs00936234_m1 (TBP). Droplets were prepared with a QX100 droplet generator and then transferred to a C1000 Touch thermal cycler (Bio-Rad Laboratories, USA). Droplet containing mixtures were analysed on a QX100 droplet reader using the Quantasoft software (Košice, Slovakia) to determine concentrations of the target DNA in copies per µl from the fraction of positive reactions using Poisson distribution analysis. Data were collected independently for each target and TBP and are presented as the ratio between target and TBP gene copies/µL. Each individual cDNA sample was analysed with technical triplicates.

### 2.5. Protein Analysis by Automated Capillary Western Blotting

Cell surface proteins were labelled with a sulfo-NHS-SS-Biotin reagent using a protein labelling kit (Pierce, Thermo Fisher Scientific). Cells were washed three times with PBS containing calcium and magnesium (PBS+) and 1 mL of biotinylation reagent (0.25 mg/mL) was added to the apical side of the filter and 1.5 mL of PBS+ to the basolateral side. The plate was gently rotated on ice at 4 °C for 30 min. The reaction was stopped by adding 50 µL of quenching buffer to the apical side and 100 µL to the basolateral side then rotating for 5 min at 4 °C. The wells were washed twice with 150 mM sodium chloride and 20 mM Trizma base, pH 7.4 and lysed with 0.5 mL of lysis buffer containing 60 mM octyl β-D-glucopyranoside, 150 mM sodium chloride, 20 mM Tris pH 7.4 and 0.5% protease inhibitor cocktail. The lysate was transferred to a microcentrifuge tube and kept on ice for 10 min with occasional vortexing before centrifugation at 14,000× *g* for 10 min at 4 °C. The supernatant was removed and saved, and a fraction containing 0.3 mg of protein (measured by the bicinchoninic acid protein assay) in 250 µL lysis buffer was added to 50 µL of streptavidin agarose resin slurry that was pre-washed with PBS+ and rotated for 1 h at room temperature. The resin was then washed, and the bound surface proteins were eluted by the addition of 40 µL of 50 mM dithiothreitol in 50% (*v*/*v*) NuPage LDS sample buffer and 50% (*v*/*v*) MilliQ water with heating for 15 min at 37 °C. Both the total cell lysate and fraction containing purified cell surface proteins were analysed for SI protein content using the 66–440 kDa separation module on an automated capillary western blotting system (‘*WES*’ ProteinSimple, Bio-Techne, UK) according to the manufacturer’s protocol. Samples were denatured for 5 min at 95 °C. The primary antibody for SI (HBB2/614/88-s, deposited by Hauri, H.-P. in Developmental Studies Hybridoma Bank, Iowa, USA) and the loading control α-actinin (MAB8279, R&D Systems, Bio-Techne, UK) were analysed in the same capillary following optimisation. A total protein concentration of 0.1 mg/mL was used for cell lysate samples and a 5-fold dilution of cell surface fractions in order to fall within the established analytical linear range. Selected samples were also digested with rapid PNGase F (New England Biolabs, Hitchin, UK) according to the manufacturer’s instructions and 5.6 µL of the digestion product was added to 1.2 µL of WES master mix and analysed for SI and α-actinin as described.

### 2.6. Measurement of Caco-2/TC7 Transport 

Caco-2/TC7 cells were cultured on 12-well transwell plates for 21 days, washed three times with warm transport buffer (20 mM HEPES buffer, 5.4 mM potassium chloride, 137 mM sodium chloride, 2.4 mM calcium chloride and 26.8 mM of sodium bicarbonate at pH 7.4) and incubated at 37 °C, 10% CO_2_ in transport buffer for 30 min. The trans-epithelial electric resistance (TEER) was measured using a Millipore Millicell Voltohmmeter (Merck Millipore, UK) at three places per well. TEER values greater than 200 MΩ indicated tight Caco-2/TC7 monolayer formation. For transport experiments, 1 mL of transport buffer was added to the basolateral side and 0.5 mL of transport solution containing glucose or sucrose, at concentrations stated in the figure legends, was added to the apical side. Samples were incubated for the specified time at 37 °C, 10% CO_2_. The apical and basolateral solutions were then transferred to microcentrifuge tubes and stored at −80 °C until analysis. During the transport assay, OLE was absent, irrespective of whether the cells had been previously treated with OLE or control.

### 2.7. Quantification of Glucose and Fructose by High Performance Anion Exchange Chromatography with Pulsed Amperometric Detection (HPAE-PAD)

The glucose and fructose content in the apical and basolateral sides after transport experiments were quantified by high performance anion exchange chromatography with pulsed amperometric detection (HPAE-PAD) using an ICS4000 system (Thermo Fisher Scientific, USA) as previously described [18].

### 2.8. Human Intervention Study

A 3-week randomised, double-blinded, placebo-controlled, crossover trial was performed. Each participant completed two seven-day treatment periods (placebo and OLE) with a week of washout in-between. The study was approved by the University of Leeds, Faculties of Mathematics and Physical Sciences and Engineering Ethics Committee (MEEC15-044). Written consent was obtained from all participants and the study was conducted between October and December 2017. To clarify, this study was separate from the study we have published previously where the OLE treatment was given acutely together with the sucrose dose. Here, the OLE was given for 1 week, and then the sucrose was given alone without the OLE, to test for indirect effects of the OLE on e.g. gene expression, SI brush border activity and protein in vivo.

Eleven female volunteers aged between 18–65 year, BMI 20–25 kg/m^2^, non-diabetic, not pregnant or lactating, not on a special diet, and not on medication, were recruited within the campus of the University of Leeds. The suitability of volunteers was evaluated by a pre-study questionnaire and a preliminary session, where volunteers were required to arrive fasted overnight and for at least 10 h. The fasting blood glucose of volunteers was measured before 10 am, and participation was eligible for volunteers with fasting blood glucose levels between 4.0 and 5.9 mmol/L. Blood glucose was analysed using an Accu-Chek Aviva blood glucose meter as previously described [13]. Subjects were randomised to placebo (control) or OLE capsules and assigned a code each. All data was stored in anonymised form and responses were unblinded at the end of the study by an external party. Capsules given to participants contained 380 mg maltodextrin per capsule (placebo) or 250 mg olecol (19.8 ± 0.5% oleuropein, equivalent to ~50 mg oleuropein) with 130 mg maltodextrin (OLE). Both placebo and OLE capsules were identical in appearance, size, and weight, and odourless. The participants were instructed to take three capsules per day, each with breakfast, lunch, and dinner. Participants were advised to retain their normal dietary habits, but were asked to consume the same identical evening meal before each measurement visit, with no olives or other olive-containing foods allowed during the study period. The final capsule of each treatment was consumed at 7 pm on the day before the visit. Participants arrived at the School of Food Science and Nutrition (University of Leeds) before 10 am in a fasted state (overnight fast of 10–12 h) for the 2 visits, and each visit lasted ~3 h. During the fasting period only water was allowed. At each visit, participants consumed a sucrose solution, prepared by dissolving 25 g white sugar in 100 mL water. Blood glucose was measured as above at 8 time points following a baseline measurement (time 0, pre-administration of sucrose solution), 15, 30, 45, 60, 90, 120, 150 and 180 min.

### 2.9. Data Analysis

Transport experiments, gene expression analysis and enzyme assays were performed using 3 independent biological replicates with 3-6 technical replicates per biological replicate. The results were normalised for each experiment by dividing all values within the experiment by the mean of the glucose control in that experiment. The mean and standard error of the mean (SEM) were determined from the normalised results of all experiments. Each gene expression sample was analysed in triplicate as per manufacturer’s recommendation. For protein analysis, two independent biological replicates with six technical replicates each were performed, and each individual sample was analysed on WES in duplicate. All results are presented as mean values from all replicates after normalisation and the error bars indicate the SEM. A one-way analysis of variance (ANOVA) followed by Tukey’s post-hoc test for statistical significant (*p* < 0.05) was performed using SPSS (v24, IBM, Armonk, NY, USA). For the human study, a two-tailed Student’s *t-*test was used to assess changes between the placebo and OLE and a *p*-value less than 0.05 was considered as statistically significant. All data are presented as mean ± standard deviation, unless otherwise specified. 

## 3. Results

### 3.1. Effect of Chronic OLE Treatment on Gene and Protein Expression in Differentiated Caco2/TC7 Cells

Treatment of differentiated Caco-2/TC7 cells for the final 3 days of differentiation with 1.5 mg/mL OLE substantially decreased SI mRNA expression by ~78% (*p* < 0.001) for cells cultured in glucose and by 74% (*p* < 0.001) for cells cultured in sucrose. The effect was time- and concentration-dependent (Figure 1). OLE treatment also reduced GLUT2 mRNA by 55% (*p* < 0.001) when cultured in glucose and by 30% in sucrose (*p* < 0.01). SGLT1 mRNA was ~27% (*p* < 0.001) lower in cells cultured in glucose and OLE. GLUT5 gene expression appeared to be less consistent, particularly when cultured in sucrose. Nevertheless, OLE induced a substantial and statistically significant (*p* < 0.01) increase in GLUT5 mRNA when cells were cultured in sucrose. OLE treatment only modestly increased CDX2, a transcription factor involved in SI expression [19], by 16% (*p* < 0.05) when cultured in glucose but not in sucrose (Figure 2). For cells cultured in glucose, OLE treatment decreased apical cell surface biotinylatable SI by 47% (*p* = 0.01) but did not affect the total cellular amount. On the other hand, OLE did not affect biotinylatable SI when cultured in sucrose, but decreased total cellular SI by 22% (*p* = 0.004) (Figure 3). 

### 3.2. Changes in SI protein Following Treatment with OLE 

The apparent molecular weight of SI increased after culture in sucrose compared to glucose (1.2%, *p* < 0.001, *n* = 24) (Figure 3), and the K_m_ also increased from 10.7 to 12.3 mM in cells cultured in sucrose compared to glucose (*p* = 0.01), although with no change in V_max_ (Figure 4). The molecular weight of SI slightly decreased after PNGase F treatment, confirming that the protein is glycosylated, and so the protein size and K_m_ differences could be related to changes in glycosylation. 

The apparent size of the SI protein slightly increased following chronic treatment with OLE in cells cultured in sucrose (0.6%, *p* < 0.001, *n* = 24), but not in glucose (Figure 3). SI activity was 31% lower following OLE treatment in cells cultured in glucose and 26% lower in cells cultured in sucrose (*p* < 0.001), and this change was concentration-dependent and occurred rapidly (Figure 4). For glucose- and sucrose-cultured cells, OLE treatment reduced the V_max_ by 34% (*p* < 0.001) and 20% (*p* < 0.001), respectively, and the K_m_ decreased from 12.3 to 9.0 mM (*p* < 0.001) with chronic OLE treatment in sucrose grown cells (Figure 4). 

### 3.3. Effect of OLE Treatment on Sucrose Hydrolysis and Transport Across Differentiated Caco-2/TC7 Monolayers 

When sucrose was added on the apical side of differentiated Caco-2 cell monolayers, both glucose and fructose time-dependently increased, demonstrating apical SI action. Furthermore, basolateral glucose and fructose increased over time (Figure 5). Changes in apical glucose and fructose, and in basolateral glucose, were sucrose concentration-dependent. Basolateral fructose was close to the limit of detection and could not be reliably measured (Figure 5). Following treatment with OLE on the apical side, there was a 38% decrease in glucose (*p* < 0.001) and a 21% decrease in fructose (*p* < 0.001) concentrations on the apical side indicating attenuation of SI activity, but no change in basolateral sugars could be detected (Figure 5). 

### 3.4. Human Intervention Study

Volunteers (11 healthy females) were 18- to 22-year-old (20.4 ± 1.29 year with average BMI 21.1 ± 3.49 kg/m^2^). After 1 week of placebo or OLE treatment, the individual responses of the subjects after ingesting sucrose solution were tested by measuring blood glucose levels (Table 1). No significant differences in iAUC, Gluc_max_ or T_max_ were observed between the two treatments in each participant. The only difference was a higher plasma glucose at 60 min in the OLE group (Figure 6).

## 4. Discussion

Following meal ingestion, sucrose hydrolysis by α-glucosidases and transport of resulting sugars across the intestinal wall are critical steps affecting post-prandial glucose concentration in the blood and present a target for interventions to blunt blood glucose spikes. SI is the only mammalian enzyme capable of hydrolysing sucrose [20], while the products, glucose and fructose, are transported across the gut wall by GLUT 2 (both sugars), GLUT5 (fructose only) and SGLT1 (glucose only) [21,22] (Figure 7). 

Enterocytes have a life-span of 3–5 days and are exposed to the contents of the gut lumen throughout differentiation while migrating from the crypt to villus tip [23]. Caco-2/TC7 cells are a well-established model for transport studies across the intestinal wall and express SI abundantly, which is modulated by sugars [24]. SI undergoes both N- and O-glycosylation which is essential for its trafficking to the apical surface and subsequent digestive activity [25]. Here we have utilized Caco-2/TC7 differentiated cell monolayers as a model to simulate the intestinal barrier and study effects of its chronic exposure to an OLE. We found that chronic OLE treatment dramatically reduced SI mRNA expression in differentiated Caco-2/TC7 cells while it also decreased SI protein and activity. Sugar transporters were also affected; when grown in sucrose, OLE lowered the mRNA expression of GLUT2, but increased expression of GLUT5, while in glucose, OLE caused a decrease in GLUT2 and SGLT1 mRNA. CDX2, one of the transcription factors required for SI expression along with HNF-1α and GATA-4 [26], modulates SI mRNA [27], but here, only a small increase or no change in CDX2 mRNA was observed, depending on growth sugar, implying that the OLE effect is through a different mechanism. In cells maintained in glucose, OLE led to a reduction in apical SI protein, accompanied by changes in kinetic properties, whereas in sucrose, OLE caused a reduction in total cellular SI protein and changes in glycosylation. Interestingly, altered glycosylation of SI had been reported in the diabetic Biobreed Wistar rats [28]. 

Based on these in vitro data, we conducted a crossover human intervention study on volunteers who consumed 50 mg oleuropein or placebo 3 times a day for 1 week, followed by a sucrose tolerance test at the end of the week. In vitro, the effect on sucrase was highly significant at 0.5 mg/mL OLE which contains 387 μM oleuropein. For the human intervention study, capsules with OLE contained 50 mg oleuropein, which, if taken with a drink of 100 mL, would be 930 μM oleuropein. Obviously the contents of the capsule would be further diluted by intestinal secretions and by food, but the concentration in the intestine could approach that of the in vitro study. It is of course difficult to define and compare exactly the in vitro and in vivo doses, and we assume complete dissolution and dispersion of oleuropein from the capsules in stomach conditions as we have shown previously [13]. The dose chosen was comparable to the dose used for 12 months in a study which showed no side effects [12]. Despite the changes observed in vitro, no apparent effects were seen in volunteers. This could be for several reasons, since this study was a pilot on a limited number of volunteers and so cannot be considered definitive. However, it does demonstrate the need to confirm in vitro data by conducting intervention studies. It is also possible that an effect could be observed in older or metabolically compromised volunteers, or for a longer treatment period, since previously, a sucrose-rich diet for 10 weeks elevated postprandial glycaemia [29]. Given the much shorter half-life of enterocytes in vivo when compared to the in vitro setting, and the significant transcriptional changes observed in vitro, we assume that SI protein turnover is slow and that SI is at a sufficiently high level in vivo that makes it more difficult to observe a change in the current intervention study. A different design, such as a more saturating dose of sucrose or a concomitant sucrose-induced chronic stress, may have led to a different outcome. 

Caco-2 cells have proven useful for investigating mechanisms, such as the involvement of lactase in the hydrolysis of flavonoid glycosides [30], where the activity was subsequently proven in vivo [31]. Here, it is possible that attenuation or modification of sucrase protein could be occurring in volunteers, but the effect was not enough to observe a measurable change, since although a small decrease in SI could have occurred, it would not have been sufficient to change the blood glucose spike if SI is not the rate-limiting step for sucrose absorption. We only used healthy young women in this pilot study and so it may also be informative to test the effect of OLE on males, or on compromised individuals such as pre-diabetic or obese volunteers. 

A very small amount of sucrose is absorbed intact without hydrolysis [32] and leads to about 0.04% of the dose in urine after a high sucrose dose of 130 g. Although oleuropein is absorbed intact to a certain extent, metabolites such as hydroxytyrosol sulfate and glucuronide are in plasma at higher concentrations [14]. However, we did not measure these in plasma as the hypothesis tested here only required interaction with the intestinal cells and not absorption into the bloodstream per se. In other studies, OLE has been used as an anti-diabetic agent [33], improved several bone biomarkers in postmenopausal women with osteopenia [12], decreased HbA1c in diabetic patients [34], improved insulin sensitivity in overweight men [16], and blunted acutely post-prandial glucose spikes in the blood when consumed together with sucrose [13]. An improvement in insulin sensitivity of 15% after 12 weeks of OLE supplementation was observed in overweight men, but this study involved an oral glucose tolerance test, not with sucrose, and so is quite different to the study presented here [16]. In another example of phenolic-induced gene expression changes, a 16 h treatment with anthocyanin-rich berry extract on Caco-2/TC7 cells led to a significant decrease in GLUT2 mRNA and GLUT2 protein and SGLT1 mRNA [35], but the consequence in vivo was not tested.

## 5. Conclusions

In conclusion, it cannot be assumed that gene expression changes will always be translatable into in vivo studies, and shows that any changes or effects by nutrients, bioactive compounds in vitro should also be validated in vivo. 

## Figures and Tables

**Figure 1 nutrients-11-01505-f001:**
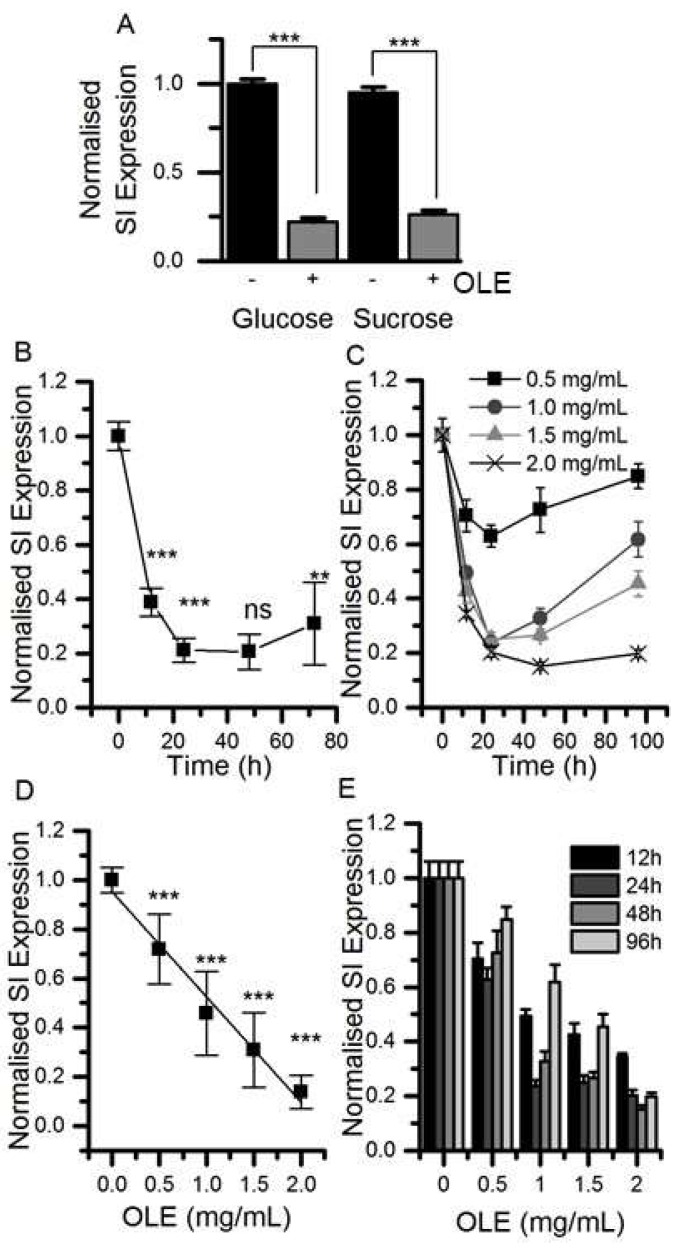
The effect of olive leaf extract (OLE) on sucrase-isomaltase (SI) mRNA expression. (Panel **A**) Differentiated Caco-2/TC7 cells were cultured in 25 mM glucose or 25 mM sucrose for 21 days and treated with 1.5 mg/mL OLE for the final 3 days of differentiation. (Panel **B**) shows the time course with 1.5 mg/mL OLE and (Panel **C**) shows the concentration-dependence, both with *n* = 9, analysed in triplicate. Results are presented as normalised mean ± SD, *p*-values determined from ANOVA followed by Tukey’s post-hoc test, labels compare to previous point, *** *p* < 0.001, ** *p* < 0.01, ns = not significant. The time course was evaluated at four OLE concentrations (**D**) and the concentration response (**E**) was evaluated at four time points (3 replicates each). SI expression is normalised to for TBP (TATA-box binding protein) as detailed in the experimental section; + indicates with OLE, - indicates without.

**Figure 2 nutrients-11-01505-f002:**
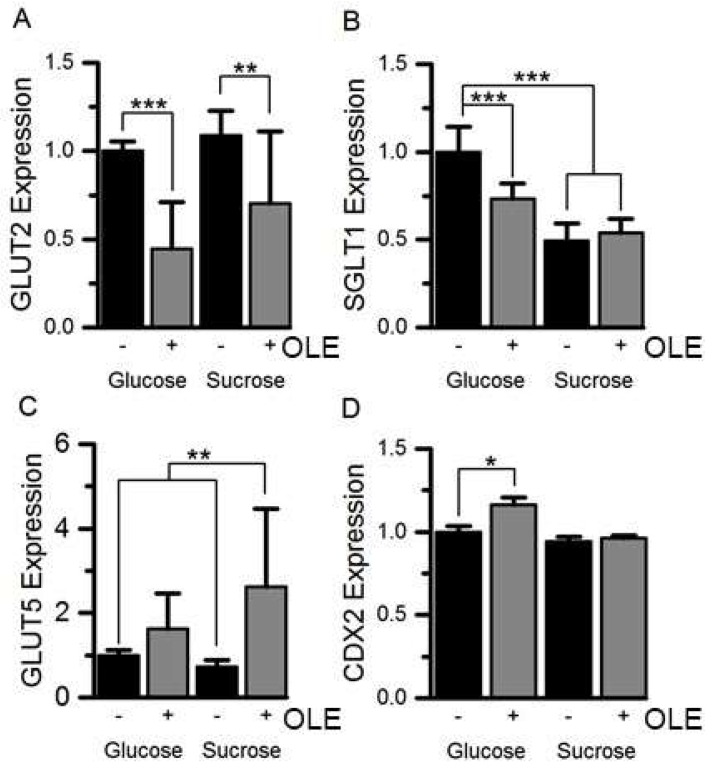
The effect of OLE on GLUT2, SGLT1, GLUT5 and CDX2 mRNA expression. GLUT2 (panel **A**), SGLT1 (panel **B**), GLUT5 (panel **C**) and CDX2 (panel **D**) mRNA expression in Caco-2/TC7 cells, cultured in 25 mM glucose or 25 mM sucrose for 21 days and treated with 1.5 mg/mL OLE for the final 3 days, was measured relative to TBP. Results are normalised to the glucose control and presented as mean ± SEM, *** *p* < 0.001, ** *p* < 0.01, * *p* < 0.05; *n* = 9, with 3 technical replicates; + indicates with OLE, - indicates without.

**Figure 3 nutrients-11-01505-f003:**
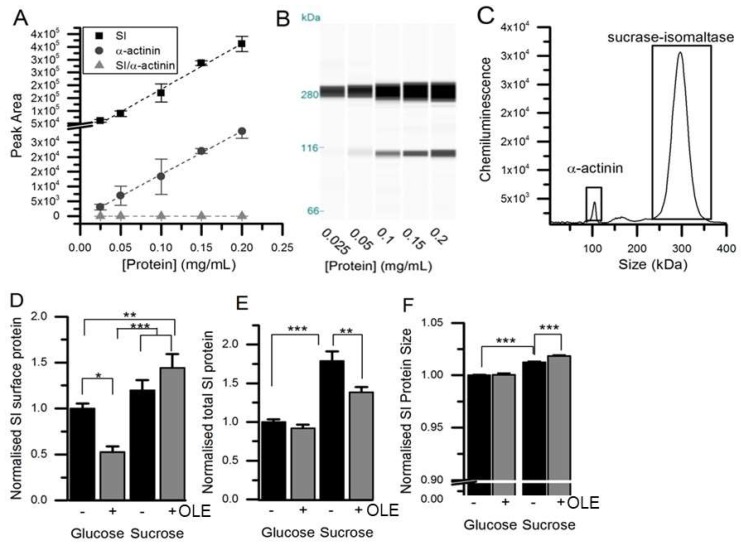
Detection and quantification of SI protein. (Panels **A**) and (Panels **B**) show the concentration-dependence of SI and α-actinin signals (Panel **C**) using automated capillary Western blotting of Caco-2/TC7 lysates. (Panel **D**) shows Caco-2/TC7 cell surface protein fraction purified after biotinylation from cells cultured on 6-well Transwell filters in glucose or sucrose for 21 days and treated on the apical side for the final 3 days with 1.5 mg/mL OLE. (Panel **E**) shows the total cellular SI, relative to α-actinin with data normalised to the glucose control and presented as mean ± SEM with *n* = 12, analysed in duplicate runs. The size was determined relative to a protein standard ladder for SI in the total lysate relative to α-actinin (**F**), normalised and presented as mean ± SEM with *n* = 24. * *p* < 0.05, ** *p* < 0.01, *** *p* < 0.001 from ANOVA followed by Tukey’s post-hoc test. There was no change in the α-actinin loading control between the groups.

**Figure 4 nutrients-11-01505-f004:**
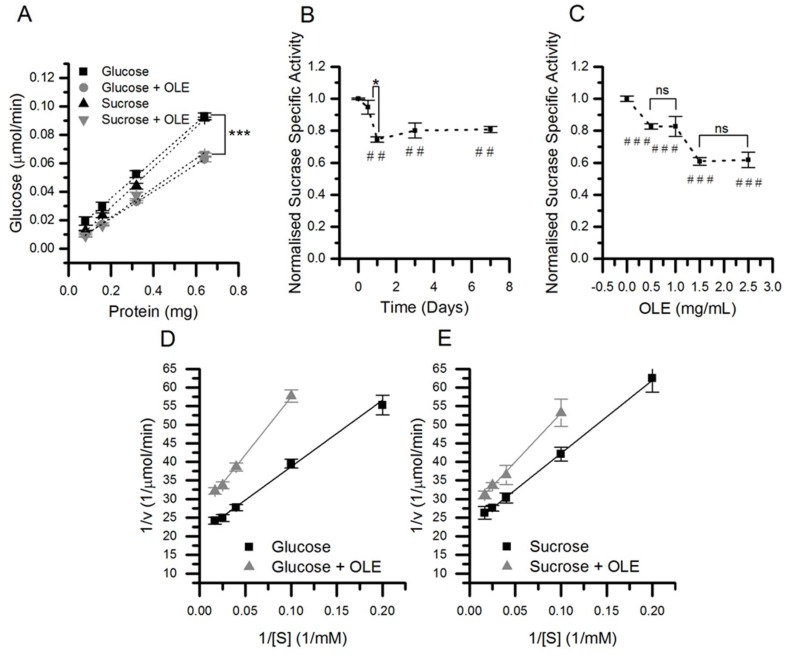
The effect of chronic OLE on sucrase enzyme kinetics. Caco-2/TC7 cells were cultured in 25 mM glucose or 25 mM sucrose for 21 days and treated for the final 3 days with 1.5 mg/mL OLE. Sucrase specific activity was determined (**A**) and the linear slope represents the enzyme activity in µmol/min.mg (*n* = 9, *** *p* < 0.001). The sucrase specific activity was evaluated up to 7 days (**B**) and at 0.5, 1, 1.5 and 2.5 mg/mL OLE, ^##^
*p* <0.01. (**C**). (*n* = 6, *p* <0.001 compared to control, * *p* < 0.05, ^###^
*p* <0.001, ns = not significant). The effect on K_m_ and V_max_ after 3 days treatment at 1.5 mg/mL was determined by Lineweaver–Burk plots for cells cultured in glucose (**D**) and sucrose (**E**) (*n* = 9). All data presented as mean ± SEM.

**Figure 5 nutrients-11-01505-f005:**
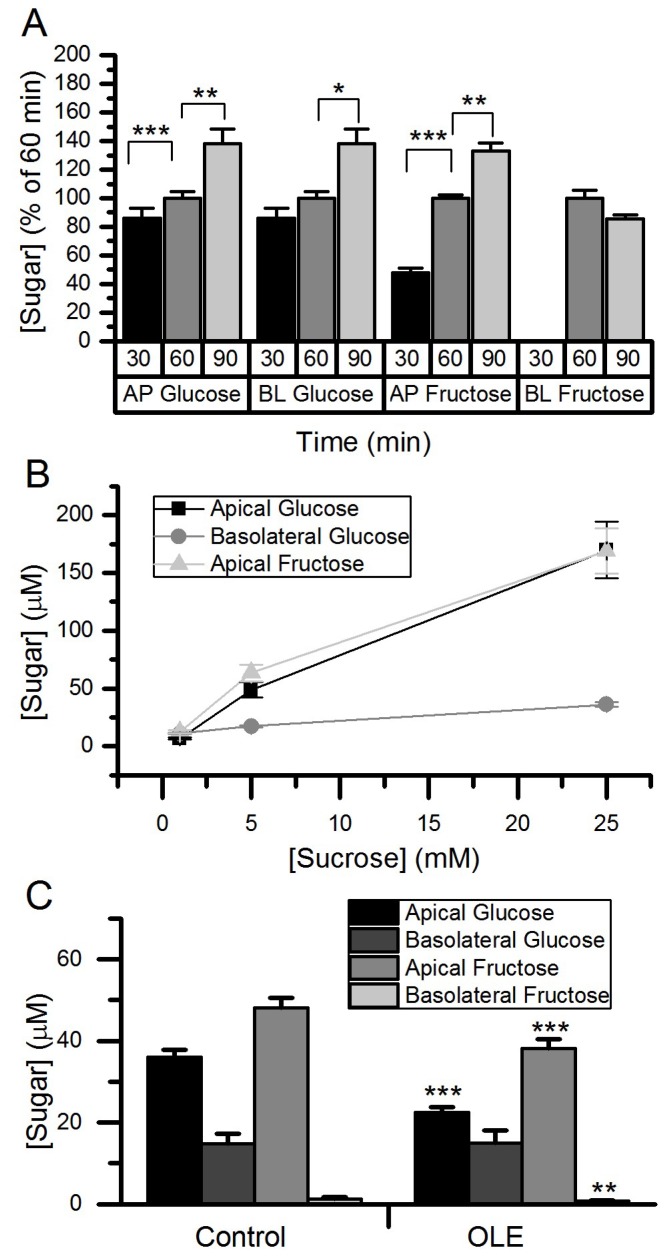
Hydrolysis and transport of sucrose in Caco-2/TC7 cells. A time course for sucrose transport using 5 mM sucrose (**A**), and transport with 1, 5 and 25 mM sucrose for 60 min (**B**). Transport with 5 mM sucrose and 60 min after chronic OLE treatment at 1.5 mg/mL for final three days of differentiation (**C**). Apical and basolateral glucose concentrations were determined by high performance anion exchange chromatography with pulsed amperometric detection (HPAE-PAD). Normalised results are presented as mean ± SEM, *n* = 18, * *p* < 0.05, ** *p* < 0.01, *** *p* < 0.001. AP = apical; BL = basolateral.

**Figure 6 nutrients-11-01505-f006:**
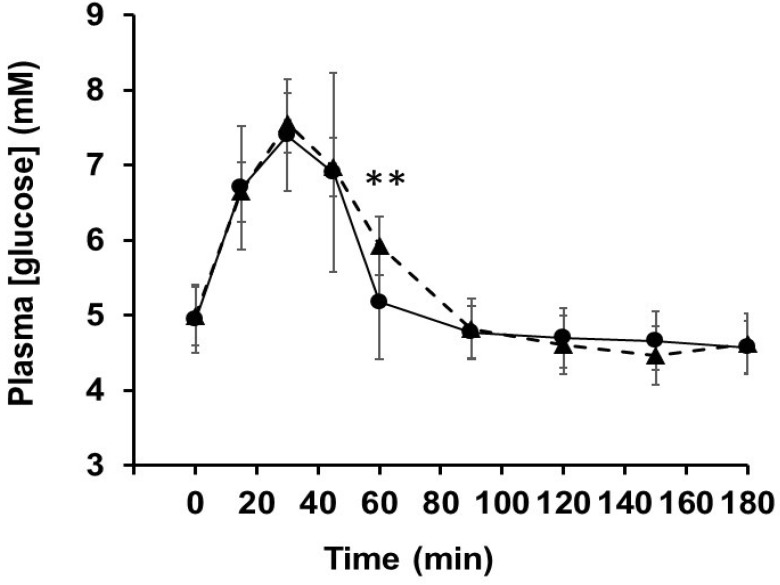
Post-prandial blood glucose concentrations. Blood glucose levels were measured in volunteers after consumption of sucrose solution following 1-week consumption of OLE (▲) or placebo (●). ** *p* < 0.01.

**Figure 7 nutrients-11-01505-f007:**
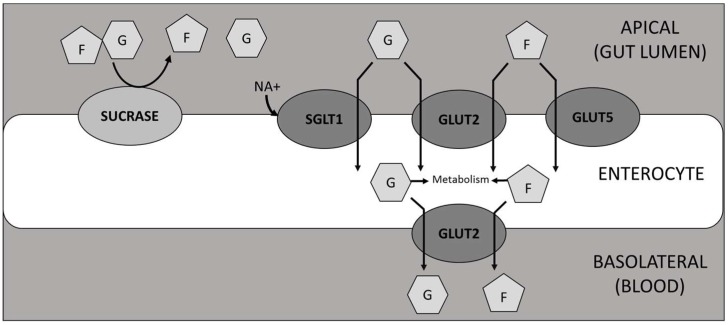
Diagrammatic representation of the hydrolysis of sucrose and subsequent transport of glucose and fructose through enterocytes. F = fructose; G = glucose.

**Table 1 nutrients-11-01505-t001:** Individual responses of the subjects after ingesting sucrose solution were tested by measuring blood glucose levels.

	Placebo	OLE	*p*-Value for Difference
Glucose_max_ (mM)	7.75 ± 0.87	7.94 ± 1.03	0.564
Glucose iAUC (min.mmol/L)	95.7 ± 15.4	112.8 ± 15.1	0.338
Fasting glucose (mM)	4.95	4.99	0.851

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
