# Peer review of "Indirect Chronic Effects of an Oleuropein-Rich Olive Leaf Extract on Sucrase-Isomaltase In Vitro and In Vivo"

_nutrients, 2019, doi:10.3390/nu11071505_

Reviewer 1 Report

The is a well written article on an interesting topic. 

The study design is appropriate and the conclusions drawn are sound. 

The discussion though needs restructuring. Your findings related to the literature should be more upfront. You should also avoid simply restating your results. 

Your human data using healthy participants do not agree with  a previous study (16) in obese individuals. A more in depth discussion as to why this is the case needs be included. 

Author Response

The discussion though needs restructuring. Your findings related to the literature should be more upfront. You should also avoid simply restating your results.

The discussion has been completely rearranged, and 2 paragraphs deleted (and the references re-ordered and renumbered).

Your human data using healthy participants do not agree with  a previous study (16) in obese individuals. A more in depth discussion as to why this is the case needs be included

A discussion on this has been added, although the study indicated by the reviewer was not involving sucrase or sucrose.

Reviewer 2 Report

This paper addressed the chronic effects of OLE on sucrose-isomaltase (SI) activity using in vitro cell models and in vivo human intervention. The topic of this study is highly relevant to the academia and industry in exploring the potential health benefits of dietary bioactives, especially related to glycemic control. The content of the manuscript is significant. And the data is clearly and comprehensively presented with scientific soundness.

The authors were able to demonstrate the effectiveness of OLE in lowering the SI activity and sucrose hydrolysis using in vitro models, but didn't observe the functional effect of OLE in controlling the blood glucose in the clinical trials. The intervention study results reported here can't be considered definitive, due to the limitation of the study design, however, it does demonstrate the need to confirm in vitro data by conducting in vivo studies. Despite the discrepancies and limitations, the authors were able to comprehensively interpret the data and draw fair conclusions out of it.

The language of this manuscript is fine but there are some formatting issues. The section and subsection numbers need to be completely checked and corrected for many sessions. Other than that, the overall merit of this manuscript is high.

Author Response

The language of this manuscript is fine but there are some formatting issues. The section and subsection numbers need to be completely checked and corrected for many sessions. Other than that, the overall merit of this manuscript is high.

The sections have been numbered correctly.